# *Arthrospira platensis* as Natural Fermentation Booster for Milk and Soy Fermented Beverages

**DOI:** 10.3390/foods9030350

**Published:** 2020-03-18

**Authors:** Francesco Martelli, Marcello Alinovi, Valentina Bernini, Monica Gatti, Elena Bancalari

**Affiliations:** Department of Food and Drug, University of Parma, 43124 Parma, Italy; francesco.martelli@studenti.unipr.it (F.M.); marcello.alinovi@studenti.unipr.it (M.A.); valentina.bernini@unipr.it (V.B.); elena.bancalari@unipr.it (E.B.)

**Keywords:** *A. platensis*, natural food additives, boosting effect, impedance analysis, commercial starter cultures, SLAB, rheological analysis

## Abstract

*Arthrospira platensis,* commercially known as Spirulina, is a fresh-water cyanobacterium that has been gaining increasing attention in recent years due to its high biological and nutritional value. For this reason, it has been employed in several food applications, to obtain or enhance functional and technological properties of cheese, yogurt, bread, cookies or pasta. The aim of this work was to evaluate the potential boosting effect of two different concentrations (0.25% and 0.50% *w*/*v*) of *A. platensis* on the fermentation capability of several starter lactic acid bacteria (LAB) strains, 1 probiotic and 4 commercial mix culture. These strains were used to ferment three different substrates and their fermentation behaviors were evaluated by impedance analyses together with rheological and color measurements. In tryptic soy broth (TSB), the *A. platensis* boosting effect was significantly higher if compared to yeast extract for all the starter LAB strains except for *Lb. casei*, which was equally stimulated. Different results were found when the same LAB strains were cultivated in SSM. The most evident boosting effect was found for *S. thermophilus* and *Lb. casei.* LAB growth was promoted by *A. platensis*, confirming that it could be a useful tool in the production of novel functional fermented dairy foods. The potential boosting effect was evaluated on four commercial mix cultures used to produce milk and soy fermented beverages. It was demonstrated that the booster effect took place, but it was variable and dependent not only on the mix culture used, but also on the substrate and *A. platensis* concentration. Also, rheological and color modifications were found to be dependent on these factors.

## 1. Introduction

*A. platensis* is a fresh-water cyanobacterium that has several biological activities and a great nutritional value, because it contains high levels of proteins (more than 60% on dry basis), essentials amino acids, vitamins, polyunsaturated fatty acids, minerals, polyphenols, carotenoids and chlorophyll [1]. Due to the presence of the pigment C-phycocyanin, the color of *A. platensis* is green-blue [2]. This microalga, commercially known as Spirulina, has different industrial applications e.g., as food supplement [3,4], in feed [5], in cosmetics [6] and in health products [7].

In several studies *A. platensis* has been employed in food formulations in order to obtain or enhance functional and technological properties of cheese [8], yogurt [9,10], soft drink beverages [11], bread, cookies and pasta [12,13,14].

Moreover, the growing demand for fermented, functional or vegan products opens new frontiers for its use as natural ingredient and/or as natural growth stimulator of starter bacteria in fermented food production. In fact, the increasing consumer’s demand of functional and natural food has led to a continuous research of new natural ingredients to be used in food formulations. [15]. Some studies reported that *A. platensis* stimulated lactic acid bacteria (LAB) growth in vitro condition [15,16,17] moreover, in milk products such as yogurt and ayran, Spirulina contributed to preservation of LAB viability during storage [18,19,20].

This aspect could be very important especially for probiotic cultures that, after ingestion, are believed to play a significant role in the intestinal tract against pathogenic microorganisms [20]. To perform this activity, a sufficient number of viable microorganisms must be present throughout the shelf life of the product [17].

Moreover, as it is well known that LAB strains have a high nutritional requirement and that their fermentation time could be influenced by numerous factors, (especially in fermented milk-related products) [21,22,23,24,25], in some cases energy boost is required in order to promote growth, fermentation capability and viability of the strains [26]. As it has been already observed, *A. platensis* biomass could increase the rate and the survival of several strains in fermented dairy products [27,28].

In this context, *A. platensis*, could play a multiple role in fermented foods: as a functional ingredient but also as a booster to enhance the technological performances of starter LAB.

The impacts of *A. platensis* on the viability of some LAB strains in fermented milk products have been already investigated by some authors [19,28] who used *A. platensis* at the end of fermentation, without considering the problems related to the microbial contaminations brought by *A. platensis.* In fact, the addition after fermentation, as in the case of yogurt at refrigeration temperatures slow down or prevent microbial cell duplication during storage.

On the other hand, as our purpose was to investigate the potentiality of *A. platensis* as fermentation booster, it was necessary to add it at the beginning of fermentation. In this case, to avoid any growth of contaminant microorganisms during fermentation that can compromise the success of fermentation and the safety of the final product, a sterilization step of *A. platensis* was necessary.

Considering all these reasons, the aim of the present research was to (i) evaluate the potentiality of sterilized *A. platensis* as natural growth booster for acidifying starter LAB strains, a probiotic strain and also commercially used acidifying starter cultures, by an innovative approach based on impedance analysis [29] (ii) study the impact of *A. platensis* addition on the overall color and rheological characteristics of the fermented milk and fermented soy-based beverage.

## 2. Materials and Methods

### 2.1. Strains and Commercial Acidifying Starter Cultures

Seven wild LAB strains, belonging to different species commonly used as acidifying starter for milk fermentation, the probiotic strain *Lactobacillus rhamnosus* GG and 4 commercial acidifying mix cultures (Mix I, II, III, and IV) were used.

The acidifying LAB strains, belonging to the collection of the Laboratory of Food Microbiology of the Department of Food and Drug (University of Parma), have been previously isolated from different food matrixes and identified by 16S rRNA sequencing (Table 1), while the commercial mix cultures were provided by Sacco Srl (Cadorago, CO, Italy).

The acidifying LAB strains and *Lb. rhamnosus* GG, maintained as frozen stock cultures in MRS (Oxoid, Ltd., Basingstoke, United Kingdom) (Lactobacillus), or M17 (Oxoid, Ltd., Basingstoke, United Kingdom) (*Lactococcus* and *Streptococcus*) broth containing 20% (*v*/*v*) glycerol at −80 °C, were recovered in Tryptic Soy Broth (TSB) (Oxoid, Ltd., Basingstoke, United Kingdom) by two overnight sub-culturing (2% *v*/*v*) at 42 °C for *Lactobacillus* and *Streptococcus*, and 30 °C for *Lactococcus* (Table 1).

Mix I, II, III and IV were maintained in lyophilized form at 4 °C until use.

### 2.2. Experimental Design

The potential boosting effect of *A. platensis* on all the chosen strains and the commercial mix acidifying starter cultures was evaluated following the experimental design reported in Figure 1. Three different growth substrates were considered: Tryptic Soy Broth (TSB) (Oxoid), reconstituted skim milk powder (SSM) (Oxoid, Ltd., Basingstoke, United Kingdom), and a commercial soy-based beverage (SBB).

TSB, reconstituted to 30 g/L (*w*/*v*), was sterilized at 121 °C for 20 min. SSM, reconstituted to 10% (*w*/*v*), was sterilized at 100 °C for 10 min. SBB, composed by water, hulled soya beans (8%), sugar, calcium carbonate, acidity regulator (Potassium phosphates), flavouring, salt, stabiliser (Gellan gum), vitamins (B2, B12, D2), iodine (Potassium Iodide), was aseptically opened and immediately used.

The dehydrated organic *A. platensis*, kindly provided by S.A.Ba.R (Novellara, RE, Italy), was rehydrated in water (5% *w*/*v*) and then sterilized at 121 °C for 20 min.

Sterilized *A. platensis* (SP) (5%) was added at 0.25% and 0.50% (*w*/*v*) to each substrate (Figure 1).

Yeast extract (Oxoid) (YE) a common LAB growth stimulant, was evaluated as positive control (C+). It was rehydrated in water (5% *w*/*v*), sterilized at 121 °C for 20 min and added at 0.25% and 0.50% (*w*/*v*) to each substrate (Figure 1).

### 2.3. Set up of Fermentation Conditions of LAB Strains

The acidifying and growth performances of LAB strains and *Lactobacillus rhamnosus* GG (Table 1) were evaluated in TSB and SSM. For TSB, the last sub-culturing step of each bacterial culture was tenfold diluted in Ringer solution (Oxoid) and inoculated (2% *v*/*v*) in 18 mL of the growth media supplemented, respectively and separately, with 0.25% and 0.50% of SP or YE (Figure 1). A negative control, without any addition of YE and SP, was also considered.

For the evaluation of acidifying and growth performances in SSM, other 24 h sub-culturing (2% v/v) of each strain in SSM were performed before use.

The inoculated media were transferred into sterilized BacTrac 4300^®^ vials (SY-LAB, Neupurkersdorf, Austria), 6 mL in each vial, and incubated at the optimal growth temperature of each strain (Table 1). All the fermentations were carried out in triplicate and monitored for 48 h by measuring the impedometric signal every 10 min, by BacTrac 4300^®^ Microbiological Analyzer (SY-LAB, Neupurkersdorf, Austria).

### 2.4. Commercial Mix Culture Fermentation

The four commercial mix bacterial cultures (Mix I, II, III and IV) (Table 1) were directly inoculated in SSM and in SBB according to the manufacturer’s instructions. As described above, the inoculated SSM and SBB were transferred into sterilized BacTrac 4300^®^, 6 mL in triplicate, and incubated at 37 °C.

All the fermentations were monitored for 48 h as previously described.

### 2.5. Impedance Measurements

Impedance measurements were performed by mean of BacTrac 4300^®^ Microbiological Analyzer system requiring the use of dedicate glass measuring cells (vials) with 4 electrodes. The system is based on the impedance splitting method and is able to register two specific impedance values for each single measurement: (i) the conventional conductance value (M) that corresponds to the media impedance, and (ii) capacitance value (E) which is the electrochemical double layer of the electrodes-electrolyte impedance [30]. Both these values, simultaneously recorded by the instrument, are shown as relative changes compared to a starting value and expressed as M% and E% [29,30]. The continuous plotting of the changes in E and M-values are visualized as E% and M% over time resulting in a capacitance or conductance curves.

Once aseptically filled with the samples, the sterilized vials were located into the appropriate position inside the BacTrac 4300^®^ incubators at the optimal growth temperature of each strain (Table 1). The two specific impedance values M% and E%, were measured and recorded every 10 min for 48 h [29,30].

The results of the impedometric measurements were analyzed as previously reported by Bancalari [29] and the Lag value was considered in this study as indicator of the potential booster effect given by SP during fermentation.

### 2.6. pH Measurement

pH was measured after 24 h of fermentation by means of pH meter (Beckman Instrument mod Φ350, Furlenton, CA, USA) and a glass electrode (Hamilton, Bonaduz, Switzerland).

### 2.7. Rheological characterization of fermented SSM and SBB

The flow behavior of SSM and SBB was evaluated, before and after fermentation, by means of an MCR 102 rheometer (Anton Paar, Graz, Austria) equipped with a 50 mm cone-plate geometry (1° angle). An aliquot of sample was placed on the lower plate and the upper cone of the rheometer was lowered to a fixed height of 104 μm. Prior to analyses, samples were subjected to a pre shear to diminish structural differences among samples caused by different treatments (200 s^−1^ for 60 s).

Flow behavior was tested by applying a linear ramp of shear rates from 10 to 200s^−1^. Temperature was set at 4 °C, and it was controlled by means of a Peltier device.

The flow behavior of the samples was described by the power law equation of the Ostwald de Waele model, as follows:τ = k(γ)^n^(1)

Where τ represents the shear stress (Pa), γ is the shear rate (s^−1^), and k and n are a consistency factor (Pa s^n^) and the flow behavior index, respectively.

Analyses were performed in triplicate before fermentation (t0) and after the fermentation process (t24).

### 2.8. Color Measurements of Fermented SSM and SBB

Color measurements were performed, before and after fermentation of SSM and SBB inoculated with four commercial mix starters, using a CR-2600d spectrophotometer (Minolta Co., Osaka, Japan) equipped with a standard illuminant D65. The instrument was calibrated prior to each analysis using a white color tile standard. An aliquot of sample (10 mL) was transferred in a 55-mm petri dish, and colorimetric measurements were performed by placing the spectrophotometer lens in direct contact with the bottom of the petri dish. Lightness of color (L* that ranges between 100 of white to 0 of black), redness (a*, that ranges between +120 of red to −120 of green), yellowness (b*, that ranges between +120 of yellow to −120 of blue) were measured in SCI mode by considering the CIE L*a*b* color space. Analysis was performed in quintuplicate in random points of the bottom surface of the petri dish. According to rheological characterization, color was measured before fermentation (t0) and after the fermentation process (t24).

### 2.9. Statistical Analysis

To investigate the effect of concentrations (0, 0.25, 0.50% *w*/*v*) of *A. platensis* and the effect of substrate (SSM and SBB), a two-way Analysis of Variance (ANOVA ) model was performed using PRC GLM of SAS (SAS Inst. Inc., NC, USA). In the case of pH, color coordinates and rheological parameters, the fermentation time was considered for the statistical analysis (0 h, corresponding to the inoculated and non-fermented sample; 24 h, corresponding to the fermented sample) and in this case, a three-way ANOVA model was considered. Prior to statistical analyses, datasets were tested for homoscedasticity, (homogeneity of variance), and normal distribution by Levene’s test and Shapiro-Wilk test, respectively, using SPSS v.25 (IBM, New York, USA).

## 3. Results and Discussion

### 3.1. Measurement of the Boosting Effect of Sterilized A. Platensis on LAB Strains by Impedance Analysis

In this study, the potential boosting effect of *A. platensis* (SP) was evaluated by mean of impedometric method, by comparing the behavior of the acidifying starter LAB strains and *Lb. rhamnosus* GG in presence or in absence of different concentration of SP in TSB and SSM. Furthermore, the potential effect of SP has been compared to that of a common LAB growth stimulant (YE) as positive control.

Both E% and M% are recorded simultaneously from the instrument, and are used to describe the microorganism growth in response to their metabolism.

In particular in SSM, M% was used as it is a less sensitive measurement, as compared to E%, but enough to register the variation of the overall impedance caused by microorganisms growth. Conversely M% is not enough sensitive to register the smallest variation in TSB, leading to a potential underestimation of the measurement, for this reason E% was used [30].

Capacitance or conductance data were analyzed by using an excel add-in and the Gompertz equation, following the method previously reported by Bancalari [29]. This allowed to obtain the kinetic parameter Lag that was used in this study to describe the potential boosting effect of *A. platensis*. Lag is measured in hours, and the greater the Lag value, the bigger the time that the strains need to adapt to the growth conditions. To better highlight this effect, the Δlag was calculated as difference between the Lag mean value of the negative control (C-) and the Lag mean value measured with SP and YE addition for each strain. The bigger the value, the higher the stimulation effect of SP and/or YE on bacterial growth (Figure 2).

The high variability of Δlag values observed for the analyzed strains means that they were differently stimulated depending on the SP and YE concentration and on the medium used (Figure 2).

In Figure 2A, the results of the analysis in TSB are reported. It was possible to observe that both the stimulators (SP and YE) had a different and variable boosting effect depending on the species. The most relevant boosting effect (bigger Δlag value) was found for *St. thermophilus* who was the fastest when SP was added to TSB at both concentrations (Figure 2A). Conversely, the effect of YE was significantly lower (*p* < 0.001) and negligible.

The YE boosting effect was significantly lower (*p* < 0.001) if compared to SP also for the other starter LAB strains except for *Lb. casei* who was equally stimulated by SP and YE (Figure 2A). These results are in agreement with a previous study [16] where a stimulation of starter LAB growth promoted by *A. platensis* in synthetic media was observed. For *Lb. rhamnosus* GG the stimulator effect was dependent on the SP concentration used; the greater effect was found at 0.50% rather than 0.25% (Figure 2A). These results are in accordance with Beheshtipour and colleagues [31], who observed that *A. platensis* had a stimulatory effect on *Lb. rhamnosus* GG, by acting as a prebiotic factor, enhancing the growth of such microorganisms and also promoting acid production during fermentation [17].

Different results were found when the same LAB strains were cultivated in SSM. Out of all strains tested, *Weissella* and *Leuconostoc* were not able to grow (Figure 2B). The reason of their incapability to metabolize lactose could be found in the fact that they were isolated from sourdough, where lactose is absent [32,33] (Figure 2B).

The most evident boosting effect of SP in SSM was found for *S. thermophilus* and *Lb. casei* (Figure 2B). While *S. thermophilus* was equally stimulated by YE and SP, independently by the concentration, *Lb. casei* was more stimulated by YE. Interestingly, *Lb. casei* 4339 (Table 1) that has been proposed as a secondary culture because of its longer Lag phase [29], in this case, with the addition of SP, has grown as fast as *St. thermophilus.*

On the other hand, *Ld. bulgaricus* was equally boosted by both YE or SP (Figure 2B). Probiotic *Lb. rhamnosus* GG strain was equally stimulated by YE (both concentration) and the lowest concentration of SP (Figure 2B). The probiotic microorganisms are often added to yogurt or yogurt-like products where sometimes are in a lower number than the minimum required for a probiotic product, because they grow slowly in milk and often shows loss of viability during storage [34]. To overcome this problem and enhance the growth and survival of these bacteria, some stimulators such as microalgae have already been proposed [26,34,35,36].

Our results are generally in agreement with some researchers [37,38] who observed that LAB growth was promoted by SP, confirming that it could be a useful tool in the production of novel functional fermented dairy foods [38]. Anyway, despite some papers have already investigated the ability of *A. platensis* as growth stimulator the approach proposed in the present study was completely different. Indeed, the impedometric method enabled the quantification (in minutes) of the time saved by using *A. platensis* as growth booster for both starter strains and mix cultures, those representing a novelty of great technological interests. On the other hand, our research differs from the others available in literature, because the *A. platensis* has been sterilized before use. This was done because the powder has its own microbial contamination [39] and its addition at the beginning of a fermentation process could lead to the germination of spores or duplication of food borne pathogenic bacteria during fermentation or storage. To avoid this problem and to ensure safety of the fermented foods, it was decided to sterilize SP.

### 3.2. Impedance Analysis of Commercial Mix Cultures

The potential boosting effect was evaluated on four commercial mix cultures used to produce fermented SSM and SBB. As far as the authors are aware, no literature data are available about the addition of SP to boost microorganisms in the production of fermented soy-based beverages. For this reason, our result can be useful, even considering the growing interest in the vegetal substitutes of the fermented milk [40].

In Table 2 the Lag values are reported as mean of 3 replicates and the boosting effect is also reported as difference between the Lag value obtained by the fermentations using SP and the control (0% SP). This parameter was expressed in minutes and, if positive, a boosting effect took place. On the other hand, a negative value indicates a growth slowdown (Table 2). To further investigate the potential effect of SP on the fermentation process, also the pH of fermented beverages was measured at the end of each fermentation (Table 2).

For the mix culture I (Mix I) no significant differences (*p* > 0.05) in the parameter Lag were found when the mix was used to ferment SSM rather than SBB without SP addition, which means that the lyophilized bacterial cells need at least the same time to duplicate and start to grow, despite the different environment (Table 2). Moreover, with the addition of both concentrations of SP, no significant differences were found in SSM. Conversely, the addition of increasing concentration of SP in SBB prolonged the Lag time (growth slowdown) of approximately 35 min with 0.25% SP and 46 min with 0.50% SP (Table 2). pH measured for the SSM fermented with mix I was significantly (*p* < 0.001) lower when 0.25% of SP was used. This means that SP, despite not being able to boost the culture at the beginning of growth, probably still enhances the fermentation capability of strains (lower pH) belonging to *S. thermophilus* species (Table 2). Interestingly, a slight improvement of fermentation capability of Mix I (lower pH) was also observed when SP was added to SBB at both concentrations and to SSM with 0.50% of SP added (*p* < 0.001). These results are in agreement with those previously reported by Molnar and colleagues (2005) [41] who observed that the addition of *A. platensis* caused a reduction of pH in yogurt samples, probably due to its biomass stimulatory effect on the *Ld. bulgaricus* growth.

Comparing the Lag values of the controls of Mix II in SSM and SBB, the parameter Lag was found to be significantly higher (*p* < 0.001) in SBB rather than in SSM; thus in this case, the growth of bacterial cells was slowed down, probably due to a difficult adaptation to the environment (Table 2). Furthermore, significant differences in Lag values (*p* < 0.001) due to SP concentrations were found when the Mix II was inoculated in SSM. In fact, a boosting effect was observed especially when 0.25% of SP was used. Even if SP addition seems to not really influence the fermentation capability of Mix II, a small improvement (24 min) was observed in SSM, especially when 0.25% of SP was used. Moreover, a significant lower pH value was found when 0.50% of SP was added. These results are in agreement with Varga and colleagues [38] who observed that pH values in milk containing *A. platensis* and inoculated with the mixed culture of *S. thermophilus* and *Ld. bulgaricus* decreased to a higher extent compared to control samples.

When Mix III was used to ferment SSM and SBB, no significant differences (*p* > 0.05) in Lag values were observed. A significant (*p* < 0.001) reduction of Lag was found only when SP was added to SSM in a concentration of 0.25%. In this case, the boosting effect was approximately 14 min. On the other hand, a negative effect of both concentrations of SP on Mix III in SBB was observed, with a growth slowdown of about 78 and 101 min respectively for the lowest and highest concentrations of SP (Table 2). Mix III seem to be suitable for fermentation of soy-based beverage without the use of SP, that can be however useful to improve the fermentation time in SSM.

For the fourth mix considered (Mix IV), Lag values of the controls (0% SP) in SSM and SBB were not found to be statistically different (*p* > 0.05). SP addition has led to a decrease of Lag value in SBB that was significant (*p* < 0.001) for both concentrations. A boosting effect of 27 min was calculated in SBB when 0.25% SP was used. Therefore, mix IV seems to be a suitable starter for the fermentation of soy-based beverages. Conversely, in SSM a slowdown of 63 and 76 min was observed when 0.25% and 0.50% were respectively added (Table 2); in this case, the slowdown was also probably associated with lower growth and acidifying ability, as it is possible to observe from the higher pH values, if compared to the control, reported in Table 2.

To sum up, a diverse and variable trend was observed when the four mix cultures were used to ferment SSM and SBB. The differences depends on the mix culture, on the substrate and on the diverse concentration of SP added, highlighting that each mix culture could have a specific application depending on the intended use.

In particular, the impedometric method used to detect Lab growth allowed, firstly, to evaluate in which fermented beverage the diverse mix cultures grew faster and, secondly, to verify if the addition of SP had a boosting effect. In particular, this method allowed to observe that the mix with the best fermentation performances in SSM were Mix I and Mix III. On the other hand, Mix IV seems to be the best choice to ferment soy-based beverage when SP was added.

### 3.3. Rheological Properties of Fermented Samples

Change in rheological properties of fermented beverages can be due to different chemical and physical modifications such as pH and proteins’ conformational changes and exopolysaccharides production [42,43]. The aim of our evaluation was to assess if the addition of *A. platensis* could indirectly improve the viscosity of fermented beverages by possibly stimulating the production of thickening molecules or the acidifying capacity of LAB cultures. The flow behavior of SSM and SBB added with the two concentration of SP, was evaluated, before (t0) and after fermentation (t24) (Table 3). Power law derived rheological parameters k and n are reported in Table 3. Considering k, all the main effects, corresponding to the evaluated growth media, SP concentrations and fermentation times, were significant (*p* < 0.05). Also, n showed significant main effects (*p* < 0.05) in almost all the cases with the exception of SP concentration in the case of mix culture I and II.

In particular, the effect of the type of beverage before fermentation on rheological properties was mainly related to the presence of thickening agents in the SBB formulation (gellan gum), that generated a more viscous liquid than SSM (k = 0.033 ± 0.008 and 0.009 ± 0.004 Pa s^n^ for SBB and SSM before fermentation, respectively), also characterized by a more pseudoplastic behavior (n = 0.71 ± 0.04 and 0.81 ± 0.08 for SBB and SSM before fermentation, respectively). As expected, the fermentation process significantly improved the viscoelastic properties of all samples (*p* < 0.05). Moreover, the change in rheological properties promoted by the fermentation was significantly influenced (*p* < 0.05) by the concentration of SP and by the mix culture used.

Consistency index k decreased or did not change for all mix cultures in SSM and SBB, with the addition of 0.25% and 0.50% SP; the decrease of k can be related to a decrease of viscosity of the treatments compared to the control samples. Accordingly, flow behavior index n increased or did not change, and this can be related to a decrease of pseudoplastic behavior of the treatments in comparison with the control samples. The only exception was represented by mix III at a SP concentration of 0.25%, that showed a significant increase of viscosity and pseudoplastic behavior (increase of k index and decrease of n index). Accordingly, in the case of SBB, mix III showed a strong increase of k index and decrease of n index at both 0.25% and 0.50% SP.

Moreover, while in the case of mix I, II and IV the enhancement of viscosity in the samples was not observed, in the case of mix III low concentrations of SP (0.25%) possibly showed the potential ability to improve and stimulate the production of thickening molecules (e.g., EPS) both in SSM and in SBB.

### 3.4. Color Characteristics of Fermented Milks

Despite the original color of *A. platensis* being green-blue [2], the thermal treatment leads to a degradation of pigments changing the color change to a dark-green or brownish color [44]. Therefore, the impact of the thermally treated *A. platensis* on the overall color characteristics of the fermented SSM and SBB has been evaluated. Results of colorimetric analyses are reported in Table 4. As it is possible to observe from the reported L*, a*, b* values, the addition of SP had a drastic impact over colorimetric characteristics (*p* < 0.001, in all mix cultures). The color of SSM and SBB changed from a whitish color of the controls (higher L* value) to a more yellowish and reddish color (higher b* and a* values) as shown in Figure 3, with the addition of 0.25% and 0.50% SP, as in the case of mix culture I. Comparing the initial color of the two substrates, SBB was characterized by different L*, a*, b* values (*p* < 0.001) in all the mix cultures (data not shown) and, in particular, by a more yellowish color than SSM (Figure 3).

The fermentation significantly changed the color characteristics of the samples (Table 4), and the modification was also noticeable by direct observation (Figure 3).

In general, the fermentation caused an increase of L* values, if compared with the non-fermented sample (*p* < 0.05 in all the comparisons), and this result was in accordance with Rankin & Brewer (1998) [45]. In particular, it was possible to notice that the increase of lightness was more marked in most of the cases for samples added with 0.25% and 0.50% SP (on average, +4.1 ± 1.6%, +4.0 ± 2.1%, +2.3 ± 1.6% for sample with 0.50%, 0.25% and 0% of SP, respectively). This can be due to the breakdown of pigment molecules such as carotenoids, xantophylls, phycobiliproteins and chlorophyll [46], as a consequence of reduced stability at low pH and microbial growth [47,48,49] or to the production of molecules that can modify the colorimetric characteristics of fermented milk [45].

Considering a* and b* values, the change of both parameters did not show a consistent trend concerning fermentation for none of the considered factors. The differences among samples were possibly conditioned by the different response of the four commercial mix cultures in relation to growth substrates characteristics and the concentrations of SP added. For example, Mix III in SBB showed a significant and evident decrease of both a* and b* values when 0.25% and 0.50% of SP were used. Conversely, when Mix III was used to ferment SSM, no significant changes between fermented and non-fermented samples were observed with the only exception of b* value in 0% SP sample (Table 4). This different behavior could be related to the different growing capacity and metabolic activities of LAB strains in different media fortified with SP.

## 4. Conclusions

The results obtained in this research allow us to conclude that Spirulina, in addition to being a well-known source of beneficial compounds that can be converted into sustainable functional compound, can also be added as natural ingredient to produce fermented milk and soy beverages with the aim to boost the fermentation performances of LAB and or improve their viability in the final product.

Considering that this type of use requires its sterilization, the results obtained in our study take on great technological relevance also because of the innovative impedometric method used. This allowed us to specifically estimate the impact of the addition of different concentration of sterilized SP on the fermentation’s behaviors of diverse acidifying LAB, probiotic strain and commercially used mix cultures.

To a different extent, the strains and commercial mix starter cultures were stimulated by the addition of SP. The encountered variability in the boosting effect of *A. platensis* was dependent on the mix culture, the substrate used, and on the diverse concentration added. This variability was also reflected on the rheological and color characteristics of fermented milks, whose change was dependent of the different growth ability and microbial specificity of LAB cultures, highlighting that each of them could have a specific employment.

Regarding the concentration of SP used, the lowest one (0.25%) showed the best boosting effect on the strains and mix culture. On the other hand, the highest concentration used (0.5%) showed an inconstant effect on LAB, displaying in some cases a stimulatory effect, or no effect or even a slowdown of LAB growth.

Knowing this variability, the SP effect should be studied depending on the strain, culture and their application, to find the right balance between the technological advantage in terms of decreasing the production times and the functional effect of spirulina addition.

## Figures and Tables

**Figure 1 foods-09-00350-f001:**
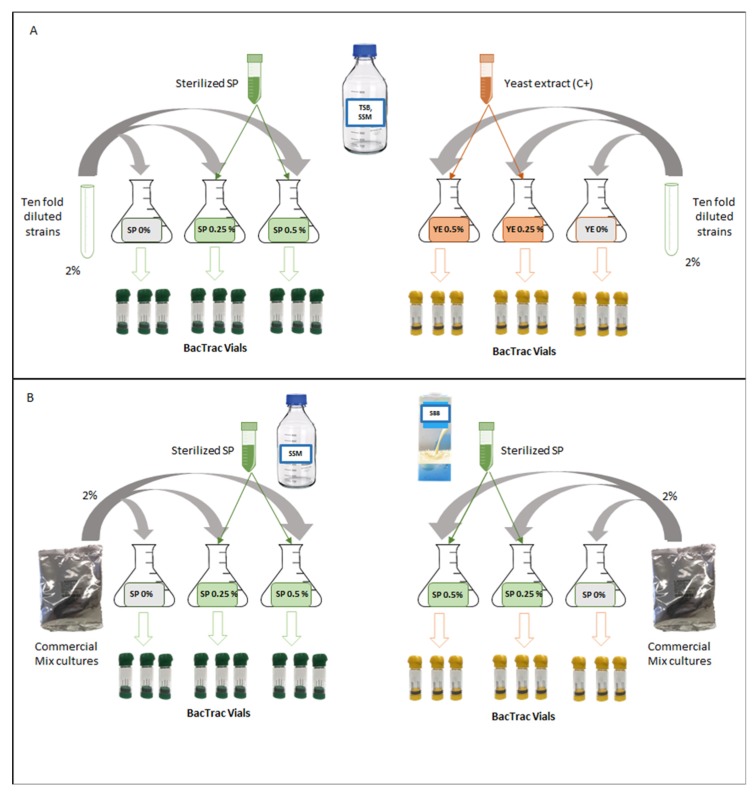
Schematic representation of the experimental design of this study: (A) Strains fermentation in TSB and SSM, (B) Commercial mix culture fermentation in SSM and SBB.

**Figure 2 foods-09-00350-f002:**
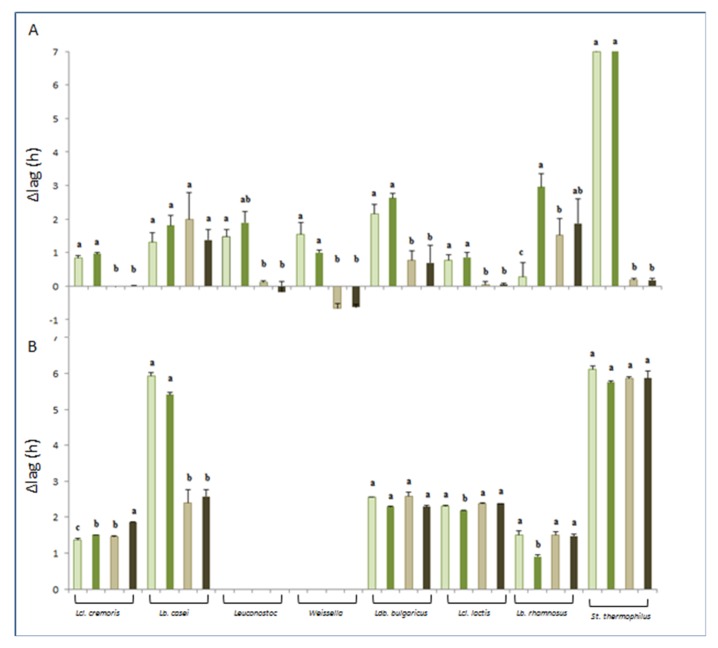
Boosting effect of *A. platensis* showed as mean Δlag valuesin Tryptic Soy Broth (TSB) (**A**) and in reconstituted skim milk powder (SSM) (**B**). ^a–b^ Different lowercase letters by column within the same mix culture indicate the presence of significant differences according to Analysis of Variance ANOVA (*p* < 0.001). Light green bars represent 0.25% of sterilized *A. platensis* (SP) and dark green 0.5%; light and dark brown bars represent respectively 0.25% and 0.5% of yeast extract (Oxoid) (YE).

**Figure 3 foods-09-00350-f003:**
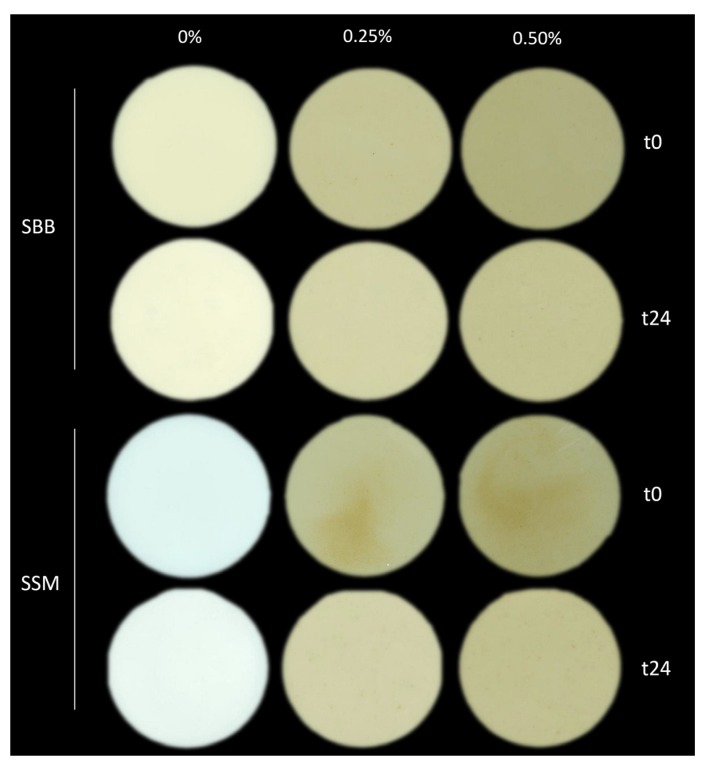
A representative picture of skimmed milk (SSM) and soy-based beverage (SBB) inoculated with commercial mix culture I, before (t0) and after 24 h (t24) of fermentation. Samples were fermented with the addition of 0, 0.25, and 0.5% (*v*/*v*) lysate of *Arthrospira platensis*.

**Table 1 foods-09-00350-t001:** Lactic acid bacteria strains (LAB) and commercial mix cultures used in this study.

Species	Strain	Abbreviation	Source	Incubation Temperature
*Lactobacillus delbrueckii subsp. bulgaricus*	2214	*Lbd. bulgaricus*	UNIPR	42 °C
*Lactobacillus casei*	4339	*Lb. casei*	UNIPR	37 °C
*Lactococcus lactis subsp. cremoris*	1978	*Lcl. cremoris*	UNIPR	30 °C
*Lactococcus lactis subsp. lactis*	2269	*Lcl. lactis*	UNIPR	30 °C
*Leuconostoc*	4456	*Leuconostoc sp.*	UNIPR	30 °C
*Weissella*	4458	*Weissella minor*	UNIPR	30 °C
*Streptococcus thermophilus*	518	*St. thermophilus*	UNIPR	42 °C
*Lactobacillus rhamnosus GG*	GG	*Lb. rhamnosus*	ATCC 53103	37 °C
*Streptococcus thermophilus multistrains*	I	*MixI*	Sacco Srl, Cadorago, Italy	37 °C
*Streptococcus thermophilus and Lactobacillus delbrueckii ssp. bulgaricus*	II	*MixII*	Sacco Srl, Cadorago, Italy	37 °C
*Streptococcus thermophilus and Lactobacillus delbrueckii ssp. bulgaricus*	III	*MixIII*	Sacco Srl, Cadorago, Italy	37 °C
*Streptococcus thermophilus, Lactobacillus delbrueckii spp. lactis and Lactobacillus helveticus.*	IV	*MixIV*	Sacco Srl, Cadorago, Italy	37 °C

UNIPR: University of Parma; ATCC: American Type Culture Collection.

**Table 2 foods-09-00350-t002:** Boosting effect of the three concentrations of *A. platensis* on mix culture.

Mix Cultures	SP%	Lag ± SD (h)	Booster Effect (min)	pH ± SD after 24 h
I in SSM	0	1.74^b^ ± 0.02	-	4.2^c^ ± 0.0
	0.25	1.68^b^ ± 0.03	3.67	3.9^e^ ± 0.0
	0.50	1.73^b^ ± 0.02	0.62	4.1^d^ ± 0.0
I in SBB	0	1.59^b^ ± 0.04	-	4.5^a^ ± 0.0
	0.25	2.17^a^ ± 0.24	−35.10	4.3^b^ ± 0.0
	0.50	2.36^a^ ± 0.08	−46.29	4.3^b^ ± 0.0
II in SSM	0	2.86^b^ ± 0.01	-	4.3^a^ ± 0.0
	0.25	2.46^d^ ± 0.04	23.73	4.3^a^ ± 0.0
	0.50	2.59^c^ ± 0.06	16.25	4.1^b^ ± 0.0
II in SBB	0	5.47^a^ ± 0.01	-	3.8^d^ ± 0.0
	0.25	5.46^a^ ± 0.02	0.85	4.0^c^ ± 0.0
	0.50	5.37^a^ ± 0.05	6.32	3.8^d^ ± 0.0
III in SSM	0	1.96^c^ ± 0.16	-	4.5^a^ ± 0.0
	0.25	1.72^d^ ± 0.02	14.15	4.3^b^ ± 0.0
	0.50	1.98^c^ ± 0.06	−1.28	4.3^b^ ± 0.0
III in SBB	0	1.97^c^ ± 0.09	-	4.0^d^ ± 0.0
	0.25	3.65^a^ ± 0.08	−101.31	4.2^c^ ± 0.0
	0.50	3.27^b^ ± 0.09	−78.37	4.2^c^ ± 0.0
IV in SSM	0	1.80^c^ ± 0.04	-	4.0^a^ ± 0.0
	0.25	2.85^b^ ± 0.03	−63.02	4.0^a^ ± 0.0
	0.50	3.07^a^ ± 0.03	−76.55	4.0^a^ ± 0.0
IV in SBB	0	1.83^c^ ± 0.02	-	3.6^b^ ± 0.0
	0.25	1.38^d^ ± 0.04	27.06	3.6^b^ ± 0.0
	0.50	1.45^d^ ± 0.07	22.97	3.6^b^ ± 0.0

^a–d^ Different lowercase letters by column within the same mix culture indicate the presence of significant differences. Lag values are reported as mean of 3 replicates for each Mix cultures and concentration (0%, 0.25%, 0.50%). The boosting effect (min) is also reported for each mix culture as difference between Lag value obtained from the fermentations using SP and the control (0% SP). The pH value is reported as mean value of three replicates.

**Table 3 foods-09-00350-t003:** Results of power-law rheological parameters n and k derived from equation 1, of each mix culture inoculated in skimmed milk (SSM) and soymilk (SBB).

			k (Pa s^n^)		n	
Mix Culture	Medium	SP (%)	t0	t24	t0	t24
I	SSM	0	0.008bB ± 0.002	4.732aA ± 0.039	0.839abA ± 0.038	0.150dB ± 0.005
		0.25	0.010bB ± 0.006	2.562bcA ± 0.221	0.839abA ± 0.120	0.262abB ± 0.015
		0.50	0.008bB ± 0.001	1.777cA ± 0.110	0.862aA ± 0.033	0.270aB ± 0.014
	SBB	0	0.034aB ± <0.001	4.513aA ± 0.034	0.705cA ± 0.003	0.182cdB ± 0.021
		0.25	0.033aB ± 0.001	4.321aA ± 0.579	0.722bcA ± 0.004	0.194cdB ± 0.033
		0.50	0.039aB ± 0.008	3.092bA ± 0.716	0.704cA ± 0.022	0.22bcB ± 0.012
II	SSM	0	0.007bB ± 0.002	3.618aA ± 0.629	0.873aA ± 0.027	0.207bB ± 0.062
		0.25	0.016bB ± 0.007	1.667bA ± 0.462	0.763abA ± 0.092	0.31aB ± 0.018
		0.50	0.009bB ± 0.005	1.062bA ± 0.033	0.847aA ± 0.070	0.362aB ± 0.001
	SBB	0	0.030aB ± 0.004	3.100aA ± 0.855	0.717bA ± 0.016	0.198bB ± 0.049
		0.25	0.033aB ± 0.001	3.333aA ± 0.380	0.720bA ± 0.008	0.158bB ± 0.034
		0.50	0.038aB ± 0.001	3.529aA ± 0.084	0.712bA ± 0.008	0.124bB ± <0.001
III	SSM	0	0.009bB ± 0.001	1.325dA ± 0.225	0.757aA ± 0.008	0.520aB ± 0.033
		0.25	0.011bB ± 0.005	2.684cA ± 0.288	0.687aA ± 0.030	0.366cB ± 0.001
		0.50	0.008bB ± 0.002	1.529dA ± 0.247	0.691aA ± 0.029	0.439bB ± <0.001
	SBB	0	0.026aB ± 0.010	2.786cA ± 0.049	0.735aA ± 0.147	0.306dB ± <0.001
		0.25	0.035aB ± 0.005	7.419aA ± 1.963	0.696aA ± 0.030	0.132eB ± 0.018
		0.50	0.025aB ± 0.005	5.038bA ± 0.293	0.744aA ± 0.035	0.277dB ± 0.011
IV	SSM	0	0.012dB ± 0.004	2.455bcA ± 0.27	0.781bA ± 0.049	0.294abB ± 0.016
		0.25	0.008dB ± 0.001	1.532cA ± 0.101	0.869aA ± 0.039	0.311aB ± 0.004
		0.50	0.007dB ± <0.001	1.294cA ± 0.110	0.894aA ± 0.017	0.317aB ± 0.008
	SBB	0	0.025aB ± <0.001	4.625aA ± 1.223	0.746bcA ± 0.001	0.150dB ± 0.021
		0.25	0.035bB ± <0.001	3.223bA ± 0.156	0.702cA ± 0.014	0.220cB ± 0.039
		0.50	0.044cB ± 0.007	2.63bcA ± 0.461	0.677cA ± 0.030	0.250bcB ± 0.013

Observations were made before (t0) and after 24 h (t24) the beginning of the fermentation. ^a–c^ Different lowercase letters indicate differences in samples fermented by a mix culture (I, II, III, IV), having different growth media (SSM, SBB) and percentages of added SP% (0, 0.25% 0.50%) and compared at the same time (t0 or t24). ^A–B^ Different uppercase letters indicate differences in the same sample (row) compared before (t0) and after (t24) fermentation.

**Table 4 foods-09-00350-t004:** Results of colorimetric coordinates L*, a*, b* of each mix culture inoculated in skimmed milk (SSM) and soymilk (SBB).

			L*		a*		b*	
Mix Culture	Medium	SP (%)	t0	t24	t0	t24	t0	t24
I	SSM	0	83.72aB ± 0.11	87.73aA ± 0.07	−4.07dB ± <0.01	−2.24fA ± 0.04	1.25dB ± 0.04	4.23eA ± 0.17
		0.25	65.17dB ± 1.13	74.41cA ± 0.04	2.83aA ± 0.75	1.64cB ± 0.09	21.08aA ± 0.09	17.10cB ± 0.15
		0.50	61.48fB ± 0.18	68.47eA ± 0.05	2.69aA ± 0.09	2.68aA ± 0.02	19.70bA ± 0.30	19.81aA ± 0.14
	SBB	0	81.45bB ± 0.01	84.47bA ± 0.02	−2.73cB ± <0.01	−2.12eA ± 0.03	15.23cA ± <0.01	14.43dB ± 0.07
		0.25	70.16cB ± 0.06	73.66dA ± 0.09	0.86bB ± 0.03	1.29dA ± 0.01	18.11bB ± 0.06	18.77bA ± 0.10
		0.50	63.91eB ± 0.17	68.46eA ± 0.04	2.21aA ± 0.13	2.05bA ± 0.02	19.21bA ± 0.24	19.67aA ± 0.03
II	SSM	0	83.23aB ± 0.07	85.18aA ± 0.08	−4.01dB ± <0.01	−2.23eA ± 0.08	0.84eB ± <0.01	2.87fA ± 0.13
		0.25	67.29dB ± 0.44	74.06dA ± 0.03	2.23bA ± 0.32	1.39cA ± 0.01	17.78cA ± 1.08	16.79dA ± 0.03
		0.50	60.65fB ± 0.26	66.76fA ± 0.43	3.72aA ± 0.16	2.82aB ± 0.07	21.51aA ± 0.55	20.42aA ± 0.30
	SBB	0	79.30bB ± 0.01	84.55bA ± 0.06	−3.64dB ± 0.01	−2.17eA ± 0.03	11.70dB ± 0.03	14.75eA ± 0.11
		0.25	70.76cB ± 0.04	74.78cA ± 0.08	0.90cB ± 0.06	1.14dA ± 0.02	18.01bcB ± <0.01	18.25cA ± 0.02
		0.50	64.83eB ± 0.28	69.36eA ± 0.19	2.32bA ± 0.21	1.97bA ± 0.04	19.19bA ± 0.42	19.73bA ± 0.21
III	SSM	0	83.56aB ± <0.01	85.61aA ± 0.21	−3.83eA ± 0.01	−3.83eA ± <0.01	1.02eB ± 0.04	1.78dA ± 0.02
		0.25	69.73cB ± 0.32	71.92cA ± 0.01	1.85cA ± 0.05	1.53cA ± 0.23	17.82cA ± 0.05	17.78bA ± 0.66
		0.50	61.27fB ± 0.71	64.48eA ± 0.04	2.77bA ± 0.04	3.30aA ± 0.44	19.93aA ± <0.01	19.69aA ± 0.48
	SBB	0	81.99bB ± 0.04	82.78bA ± 0.02	−3.07dB ± 0.01	−2.66dA ± 0.01	13.30dB ± 0.08	14.54cA ± 0.12
		0.25	67.39dB ± 0.05	69.52dA ± 0.01	2.84bA ± 0.02	1.28cB ± 0.02	19.74aA ± 0.06	16.76bB ± 0.36
		0.50	61.95eB ± <0.01	63.39fA ± 0.07	3.32aA ± 0.04	2.45bB ± 0.01	19.31bA ± 0.23	17.74bB ± 0.02
IV	SSM	0	81.59aB ± 0.01	84.65aA ± 0.02	−3.48fB ± 0.02	−2.20dA ± 0.01	−1.15fB ± 0.03	2.75eA ± 0.01
		0.25	67.56cB ± 0.04	71.48dA ± 0.03	0.75dB ± 0.01	1.88bA ± 0.04	15.05dB ± <0.01	17.81cA ± 0.01
		0.50	60.17eB ± 0.04	64.75fA ± 0.17	1.80bB ± 0.02	2.97aA ± 0.04	16.92cB ± 0.11	20.88aA ± 0.08
	SBB	0	80.32bB ± 0.08	84.07bA ± 0.02	−3.41eB ± <0.01	−2.36eA ± 0.04	13.09eB ± 0.10	14.03dA ± 0.09
		0.25	67.44cB ± 0.10	72.02cA ± 0.06	0.80cB ± 0.02	1.12cA ± 0.05	17.93bA ± 0.13	17.90cA ± <0.01
		0.50	61.97dB ± 0.06	66.29eA ± 0.05	1.93aA ± 0.03	1.81bB ± 0.01	18.70aB ± 0.02	19.18bA ± 0.05

Observations were made before (t0) and after 24 h (t24) the beginning of the fermentation. ^a–f^ Different lowercase letters indicate differences in samples fermented by a mix culture (I, II, III, IV), having different growth media (SSM, SBB) and percentages of added SP% (0, 0.25% 0.50%) and compared at the same time (t0 or t24). ^A–B^ Different uppercase letters indicate differences in the same sample (row) compared before (t0) and after (t24) fermentation

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
