# Peer review of "Arthrospira platensis* as Natural Fermentation Booster for Milk and Soy Fermented Beverages"

_foods, 2020, doi:10.3390/foods9030350_

Round 1
Reviewer 1 Report
This is a resubmitted manuscript.
However, my previous comments were not taken under consideration.
Therefore, I insist in my previous comment (the line numbers is according to the first original submission).
I cannot find the novelty and originality of this work. Please see the following 3 comments:
Several applications in foods and especially dairy products. See ref 8-14. It is well known that spirulina stimulated lactic acid bacteria (LAB) growth. See ref 15-18, 26. Line 215-218. “Our results are in agreement with some researchers [35,36] who observed that LAB growth was promoted by SP, confirming that it could be a useful tool in the production, for example, of novel functional fermented dairy foods [36]. On the other hand, our research differs from the others available in literature, because the A. platensis has been sterilized before use.” The novelty is only sterilization?
Other comments:
Line 51. as far as. Correct
Lactococcus lactis subsp. Lactis 2269 has been already tested in the previous study of the authors. See ref 27
Table 1. Leuconostoc???? Weissella??? Only this???
What about “Çelekli, A., Alslibi, Z. A., & üseyin Bozkurt, H. (2019). Influence of incorporated Spirulina platensis on the growth of microflora and physicochemical properties of ayran as a functional food. Algal Research, 44, 101710.”?
Please explain the results in mix cultures. All microorganisms affected the same? Are you sure that, in mix cultures, all the microorganisms growth? Are you sure that you did not measure only for example S. thermophilus? In my opinion the results of the present study and especially those in mix culture have no meaning.
This work has not been prepared and designed properly. It seems more like 2 different works that have been combined. Works in dairy matrices are well known but probably in soy matrix would have an interest. However, the application of this method do not provide information about the growth of each microorganism, when mixed cultures are used. Microbiological analyses would be more appropriate.
Section 3.3. Rheological properties of fermented samples and 3.4. Color characteristics of fermented milks are not relevant with the aim of the study and the title.
Lines 378-380. “Regarding the concentration, the lowest one seems to have a constant boosting effect on the strains and mix culture, for this reason, further investigation will be carried out to assess the possibility to use a less amount of SP in the formulations, also considering the high cost of SP.” Please explain why the experiments with lower concentrations of SP were not included in the present study. I cannot understand how these experiments would be used in a future manuscript. This is also proves that the work has not been designed properly."
Please this time I want a reply from the authors in each comment and especially regarding similar results already mentioned in the literature.
The new version has only corrections/additions in introduction and conclusion...
Author Response
This is a resubmitted manuscript.
However, my previous comments were not taken under consideration.
Therefore, I insist in my previous comment (the line numbers is according to the first original submission).
I cannot find the novelty and originality of this work. Please see the following 3 comments:
Several applications in foods and especially dairy products. See ref 8-14. It is well known that spirulina stimulated lactic acid bacteria (LAB) growth. See ref 15-18, 26. Line 215-218. “Our results are in agreement with some researchers [35,36] who observed that LAB growth was promoted by SP, confirming that it could be a useful tool in the production, for example, of novel functional fermented dairy foods [36]. On the other hand, our research differs from the others available in literature, because the A. platensis has been sterilized before use.” The novelty is only sterilization?
We are really sorry that the reviewer couldn’t find the originality of our findings. Despite some papers have already investigated the ability of A. platensis, in particular, and algae, in general, to stimulate the growth of some LAB strains, our approach was completely different from the other available. First of all, we applied a method that enables the quantification (in minutes) of the time saved by using A. platensis as growth booster for both starter strains and mix cultures. We think that this is a novelty of great technological interest. Furthermore, we decided to use sterilized A. platensis since, as already well reported in literature, its initial contamination could compromise not only the fermentation, but also the safety of the final product; and this is another novelty point. Moreover, no studies exist on the comparison of the effect of A. platensis on different strains by using impedometric method, but also no studies exist on the effect of A. platensis on the fermentations behaviors of different mix cultures compared in milk and in soya drink.
Other comments:
- Line 51. as far as. Correct. The text has been modified accordingly, line 62-63
- Lactococcus lactis subsp. Lactis 2269 has been already tested in the previous study of the authors. See ref 27. We perfectly know that this strain was already used in a previous study but the aim was completely different.
- Table 1. Leuconostoc???? Weissella??? Only this??? Ok we have correct the formal error as follows in Table 1: Leuconostoc sp and Weissella minor. In any case, I think that is not a polite and constructive way to make suggestions, especially among colleagues
- What about “Çelekli, A., Alslibi, Z. A., & üseyin Bozkurt, H. (2019). Influence of incorporated Spirulina platensis on the growth of microflora and physicochemical properties of ayran as a functional food. Algal Research, 44, 101710.”? We didn’t cite this article because when we wrote our paper it wasn’t published yet, but we add the citation in the text [20]. Line 48 in the text. Also in this case, for this useful suggestion, could be highlighted in a more politer and constructive way, then again, we do the same work.
- Please explain the results in mix cultures. All microorganisms affected the same? Are you sure that, in mix cultures, all the microorganisms growth? Are you sure that you did not measure only for example S. thermophilus? In my opinion the results of the present study and especially those in mix culture have no meaning.
We are sorry but we cannot agree with the reviewer because this was not the aim of the paper. For sure it is well known that microorganisms in the mix cultures grow differently, but the aim was to evaluate the impact of A. platensis addition on the overall growth behavior of the mix cultures, as compared to a control without the addition of the booster. From an industrial point of view, save time means save money, and with our approach we were able to quantify such mentioned amount of time. In fact, we were able to estimate the effect of each concentration of A. platensis added, toward each mix cultures, in minutes. We are convinced that for industries, especially those who produce starters or those who work with starter cultures, this could be a very interesting topic.
- This work has not been prepared and designed properly. It seems more like 2 different works that have been combined. Works in dairy matrices are well known but probably in soy matrix would have an interest. However, the application of this method do not provide information about the growth of each microorganism, when mixed cultures are used. Microbiological analyses would be more appropriate.
We are really sorry that the reviewer think that “the work has not been prepared and designed properly”. We respect his/her opinion with whom we completely disagree.
Considering the method use for all evaluation, we are convinced that the paper has a correct and rational approach that we will try to better explain: we first decided to test some strains belonging to our collection that can be used as starters or adjunct cultures in fermentation processes. In this way we wanted to initially test the method and understand in a “simplified” condition the impact of A. platensis on the growth abilities of the strains. For this reason, we started to test the strains in their optimal growth condition (TSB). After that, as it is well known the industry’s growing interest in the topic, it was decided to test also the impact of the booster on the growth behavior of mix cultures, first of all in SSM (as these mix are commonly used in dairy products) and then in soya drink. This was done to understand if the addition of SP could, in some way, boost the growth of mix cultures and it could help those to grow in a diverse environment such as soya drink.
It is well known that mix cultures are intentionally designed and studied to allow the differential growth of the diverse species that compose the mix. Despite it could be interesting, we didn’t focus on the single-species growth behavior within each mix cultures, because this was not the aim of the research.
Finally, we are not able to understand the meaning of the sentence “Microbiological analyses would be more appropriate”. The application of impedometric method is one of the novelty of the research. Of course, the method may not please the reviewer, but anyway Impedance analysis is a microbiological analysis. The reviewer could explained the reasons or its point of view, in this case a constructive interaction with the authors would occurs. I think that this is the main task of the reviewers.
- Section 3.3. Rheological properties of fermented samples and 3.4. Color characteristics of fermented milks are not relevant with the aim of the study and the title.
We also disagree with this reviewer's opinion. Rheological and colorimetric properties are of primary importance in the case of fermented beverages because these properties can have a strong impact on the sensory evaluation and overall appreciation of a product.
As in this study we would like to investigate the boosting effect, and of course the applicability of A. platensis as an ingredient in fermented beverages, it was important also to evaluate the effect of this addition on the final characteristics of the product, and not only the behavior during fermentation. We believe that by stating that Spirulina can have an effect not only as a fermentation booster on starter cultures and strains, but also as “thickening booster” in the final fermented product (as highlighted in some samples), it can improve quality and relevance of this research.
- Lines 378-380. “Regarding the concentration, the lowest one seems to have a constant boosting effect on the strains and mix culture, for this reason, further investigation will be carried out to assess the possibility to use a less amount of SP in the formulations, also considering the high cost of SP.” Please explain why the experiments with lower concentrations of SP were not included in the present study. I cannot understand how these experiments would be used in a future manuscript. This is also proves that the work has not been designed properly."
The choice of the concentrations of A. platensis used was done based on the data available in literature. We think that the choice of the concentration to be used have to respect a balance between the high cost of A. platensis (better lower amounts) and the potential functional properties that the addition could confer to a final product (higher amount). Therefore, we only want to conclude that the perfect amount to be used depend on the application.
Please this time I want a reply from the authors in each comment and especially regarding similar results already mentioned in the literature.
The new version has only corrections/additions in introduction and conclusion...
Reviewer 2 Report
My previous comments:
This article aims to evaluate the potential boosting effect of two different concentrations of Arthrospira platensis (SP) on the fermentation capability of several species of LAB in tryptic soy broth, reconstituted skim milk, and soy-based beverage. The authors concluded that the booster effect was observed but it was variable and dependent on the strain used but also on the substrate and SP concentration. They also addressed that rheological and color modifications were also dependent on these factors. Although the article is not innovative, it contains original and interesting information. In general, please confirm the statistical analysis and their descriptions in the article. Titles for Figures 2 and 3, and Tables 1, 2 and 3 are too descriptive and need to be rephrased as titles in short (i.e., Table 2. The effect of Arthrospira platensis concentrations on lag values, booster effect, and pH in two growth substrates.) Their descriptive statements can be placed under the figures and tables as footnotes. Line 15. Revise to “A. platensis”. Lines 19-20. Revise to “A. platensis”. Lines 73-74. Shorten the title. Table 1. Describe “UNIPR” under the table as a footnote. Lines 79-80. Any justification for the selection of SSM and SBB. Line 87. Are these final concentrations (or 0.25 and 0.5% of 5% rehydrated SP) of SP to each substrate? Line 95. Why was not SBB investigated? Line 103. Revise to “triplicate”. Line 114. Rephrase to “Once aseptically filled with the samples, the sterilized vials were located ….”. Line 116. Rephrase to “The two specific impedance values M% and E% were measured and …” Lines 122-123. Revise to “pH meter (Beckman Instrument model 122 Φ350, Furlenton, CA, USA) …”. Line 142. Revise to “prior to”. Line 152. Revise to “To investigate the effect of concentrations (0, 0.25, 0.50%) of A. platensis and the effect of substrate (SSM and SBB), a two-way ANOVA …”. Lines 154-157. Revise to “In the case of pH, color coordinates and rheological parameters, the fermentation time was also considered for the statistical analysis (0h, corresponding to the inoculated and non-fermented sample; 24h, corresponding to the fermented sample) …”. Line 159. Revise to “A. platensis”. Line 160. Remove “,” after “(SP)”. Line 164. Revise to “stimulant (YE) as positive control”. Lines 165-172. Consider moving these sentences to either the “Introduction” section or “Materials and Methods” section. Line 171. Remove “changes” after “M%”. Line 173. Clarify why M% was used in SSM. Line 182. Remove “,”. Line 185. Remove “broth”. Lines 190-191. Remove “independently from SP and YE concentration”. Line 192. Revise to “Figure 2A”. Lines 199-200. Revise to “Different results were found when the same LAB strains were cultivated in SSM. Out of all strains tested, Weissella and Leuconostoc were not able to grow (Figure 2B).” Line 204. What is “the first”? Rephrase it. Lines 206-207. Resulted what? Line 215. Add “generally” between “are” and “in agreement”. Line 216. Remove “for example”. Lines 215-222. What are the authors trying to say here? Are the autors trying to say that although their research used SP, their results agree with the results of other researchers? Figure 2. Confirm statistical differences (i.e., due to SP concentration of Leuconostoc and YE concentration of Lb. casei, etc.). Besides, Figures stand alone. Therefore, clarify what A and B are for. Line 224. Make a short title and Revise to “Results of …”. Lines 227-228. Move to as a part of footnote under Figure 2. Line 239. Clarify “the pH of ? was measured …” Line 247. Remove “,”. Lines 248. Revise to “means”. Line 259. Clarify a significant difference of what? Line 265. Revise the sentence. How do authors know if it decreased rapidly? Lines 268-269. How did the authors come up with this conclusion? Lines 275-276. Authors may want to consider to run statistical analysis for SP concentrations in SBB only to come up with this conclusion. Lines 288-290. This statement doesn’t seem to apply to SBB. Lines 292-295. The title is too descriptive and needs to be rephrased as a short title and its descriptive statement can be placed under the table as a footnote. Table 2. Authors may need to be consistent with statistical analysis in alphabetical superscripts starting from a with highest values, b with the next highest values, and so forth. Line 305. Remove “,”. Line 317. Remove “the presence and”. Lines 326-330. Two sentences seem to deliver the same message. Consider to combine and shorten sentences. Lines 332-334. Title is too descriptive and need to be rephrased as a short title and its descriptive statement can be placed under the table as a foot note. What do lowercase letters (i.e., a) and uppercase letters (i.e., A) compare in statistical analysis? Revise to “soybased beverage (SBB)”. Be consistent with terms either “SoyaD” or “SBB”. Lines 337-339. Revise to “Despite the original color of A. platensis is green-blue [2], the thermal treatment leads to a degradation of pigments changing the color to a dark-green or brownish color [42]. Therefore, the impact …”. Line 344. Revise to “as shown in the Figure 3.” Line 353. Are the values of the lightness on average or one represetnataive values? Lines 361-362. Revise to “For example, Mix III in SBB showed a significant and evident decrease …”. Line 363. Add “,” after “SSM”. Line 364. Remove “,”. Line 368-371. Revise to “Figure 3. A representative picture of skimmed milk (SSM) and soy-based beverage (SBB) inoculated with commercial mix culture I , before 368 (t0) and after 24 hours (t24) of fermentation.”. Samples were fermented with the addition of 0, 0.25, and 0.5% (v/v) lysate of Arthrospira platensis. Lines 376-377. Revise to “A. platensis”, “used, and on the concentration …”. Line 379. Revise to “… mix culture. Therefore, further …”.
The revised one slightly improved the quality of the article. However, the authors have not addressed most of my comments in the revised one (i.e., titles for Figures 2 and 3, and Tables 1, 2, and 3 are too descriptive and need to be rephrased as titles in short and statistics).
Author Response
This article aims to evaluate the potential boosting effect of two different concentrations of Arthrospira platensis (SP) on the fermentation capability of several species of LAB in tryptic soy broth, reconstituted skim milk, and soy-based beverage. The authors concluded that the booster effect was observed but it was variable and dependent on the strain used but also on the substrate and SP concentration. They also addressed that rheological and color modifications were also dependent on these factors. Although the article is not innovative, it contains original and interesting information. In general, please confirm the statistical analysis and their descriptions in the article.
- Titles for Figures 2 and 3, and Tables 1, 2 and 3 are too descriptive and need to be rephrased as titles in short (i.e., Table 2. The effect of Arthrospira platensis concentrations on lag values, booster effect, and pH in two growth substrates.) Their descriptive statements can be placed under the figures and tables as footnotes.
Thank you for the suggestion. Titles have been modified accordingly.
- Line 15. Revise to “A. platensis”. Ok, revised (Line 16)
- Lines 19-20. Revise to “A. platensis”. Ok, revised (Lines 20-21)
- Lines 73-74. Shorten the title. Ok, revised (Lines 100-101)
- Table 1. Describe “UNIPR” under the table as a footnote. UNIPR: University of Parma and ATCC: American Type Culture Collection were added, lines 102-103
- Lines 79-80. Any justification for the selection of SSM and SBB. Thanks for the comment. We agree, we didn’t explain the choice of this two substrates. SSM and SSB were selected in order to study the possibility to produce two different fermented beverages. To reach that goal we needed sterile substrate. A reconstituted skim milk powder (SSM) was used to produce “fermented milk”. On the other hand, to avoid sterilizing one soy drink, a commercial soy-based beverage (SBB) was used to produce “soy fermented beverage”.
- Line 87. Are these final concentrations (or 0.25 and 0.5% of 5% rehydrated SP) of SP to each substrate? Yes, these are the final concentrations in the medium
- Line 95. Why was not SBB investigated? SBB was not investigated in this phase because we chose to study the boosting effect of platensis on single strains only in a known condition. TSB as it is a common growth medium that could be used for all the strains considered, and SSM because milk is one of the most common product used for LAB fermentations. The second step of the work, on the other hand, is much more applicative. In fact, we used 4 commercial mix starter. In this case, in addition to milk, for which the blends are normally used, we also wanted to try the soy drink.
- Line 103. Revise to “triplicate”. Ok revised, line 139
- Line 114. Rephrase to “Once aseptically filled with the samples, the sterilized vials were located ….”. ok rephrased, line 158
- Line 116. Rephrase to “The two specific impedance values M% and E% were measured and …” ok rephrased, line 160
- Lines 122-123. Revise to “pH meter (Beckman Instrument model 122 Φ350, Furlenton, CA, USA) …”. ok revised, line 166
- Line 142. Revise to “prior to”. ok revised, line 186
- Line 152. Revise to “To investigate the effect of concentrations (0, 0.25, 0.50%) of A. platensis and the effect of substrate (SSM and SBB), a two-way ANOVA …”. ok revised, line 196
- Lines 154-157. Revise to “In the case of pH, color coordinates and rheological parameters, the fermentation time was also considered for the statistical analysis (0h, corresponding to the inoculated and non-fermented sample; 24h, corresponding to the fermented sample) …”. ok revised, line 198-201
- Line 159. Revise to “A. platensis”. Ok, revised
- Line 160. Remove “,” after “(SP)”. Ok, revised (Line 204)
- Line 164. Revise to “stimulant (YE) as positive control”. Ok, revised (Line 208)
- Lines 165-172. Consider moving these sentences to either the “Introduction” section or “Materials and Methods” section. We moved the sentence to materials and methods section as suggested (Lines 150-157)
- Line 171. Remove “changes” after “M%”. Ok, removed
- Line 173. Clarify why M% was used in SSM. M% is the variation of impedance of culture medium (Zm), which is recorded as the relative change in conductivity compared to an initially recorded value, and visualized as M% change (conductance) over time. This measurement refers to the overall impedance variation in the medium, in this case milk. As reported by Mucchetti et al 1994, milk has conductive properties because it is rich in charged compounds, especially mineral salts. The electrical conductivity of milk is determined primarily by sodium and chloride ions but also by other ions. For these reasons, milk is considered as an optimal conductivity growth medium, as it maximized conductivity changes. Therefore, M% was used as is a less sensitive measurement, but enough to register a correct variation of the overall impedance caused by microorganisms’ growth inside the milk. Also E% measurement might be used, but due to its high sensitivity it could lead to an overestimation of the measurement.
- Line 182. Remove “,”. Ok, removed
- Line 185. Remove “broth”. Ok, removed (Line 230)
- Lines 190-191. Remove “independently from SP and YE concentration”. Ok, removed (Lines 235-236)
- Line 192. Revise to “Figure 2A”. Ok, revised
- Lines 199-200. Revise to “Different results were found when the same LAB strains were cultivated in SSM. Out of all strains tested, Weissella and Leuconostoc were not able to grow (Figure 2B).” Ok, revised (Lines 244-245)
- Line 204. What is “the first”? Rephrase it. Ok, we specified the subject of the sentence (line 251)
- Lines 206-207. Resulted what? We rephrased. (Line 254)
- Line 215. Add “generally” between “are” and “in agreement”. Ok, added (line 262)
- Line 216. Remove “for example”. Ok, removed (line 263)
Lines 215-222. What are the authors trying to say here? Are the autors trying to say that although their research used SP, their results agree with the results of other researchers?
Despite some papers have already investigated the ability of A. platensis, in particular, and algae, in general, to stimulate the growth of some LAB strains, our approach was completely different from the other available. First of all we applied a method that enables the quantification (in minutes) of the time saved by using A. platensis as growth booster for both starter strains and mix cultures. We think that this is a novelty of great technological interests. Furthermore, we decided to use sterilized A. platensis since, as already well reported in literature, its initial contamination could compromise not only the fermentation, but also the safety of the final product; and this is another novelty point. Moreover, no studies exist on the comparison of the effect of A. platensis on different strains by using impedometric method, but also no studies exist on the effect of A. platensis on the fermentations behaviors of different mix cultures compared in milk and in soya drink.
Considering all that, we are grateful to the reviewer for the comment which tells us how these considerations were not well highlighted. For this reason we better explained this part in the text. (Lines 264-268)
- Figure 2. Confirm statistical differences (i.e., due to SP concentration of Leuconostoc and YE concentration of Lb. casei, etc.). We confirm
- Besides, Figures stand alone. Therefore, clarify what A and B are for. A and B have been clarified in the figure caption.
- Line 224. Make a short title and Revise to “Results of …”. Ok, revised (lines 276-278)
- Lines 227-228. Move to as a part of footnote under Figure 2. Ok, revised
- Line 239. Clarify “the pH of ? was measured …” Ok, revised (line 299)
- Line 247. Remove “,”. Ok, revised
- Lines 248. Revise to “means”. Ok, revised (line 308)
- Line 259. Clarify a significant difference of what? Ok, revised. We specified that the difference was in Lag values (Line 319)
- Line 265. Revise the sentence. How do authors know if it decreased rapidly? Revised (Lines 325-326)
- Lines 268-269. How did the authors come up with this conclusion? Even if the growth of MIX III wasn’t boosted by adding SP, in the control sample the Lag phase in SBB was the same of SSM. Considering that MIX III is a yogurt starter commonly used in milk, we can consider it a good value for production.
- Lines 275-276. Authors may want to consider to run statistical analysis for SP concentrations in SBB only to come up with this conclusion. We are sorry but we are not able to understand the suggestion. Significant differences according to ANOVA have been calculated and reported in table 2
- Lines 288-290. This statement doesn’t seem to apply to SBB. Sentence revised. Lines 349-350
- Lines 292-295. The title is too descriptive and needs to be rephrased as a short title and its descriptive statement can be placed under the table as a footnote. Ok, revised (Line 353)
- Table 2. Authors may need to be consistent with statistical analysis in alphabetical superscripts starting from a with highest values, b with the next highest values, and so forth. Revised
- Line 305. Remove “,”. Ok, revised
- Line 317. Remove “the presence and”. Ok, revised (Line 384)
- Lines 326-330. Two sentences seem to deliver the same message. Consider to combine and shorten sentences. Ok, revised (Line 393)
- Lines 332-334. Title is too descriptive and need to be rephrased as a short title and its descriptive statement can be placed under the table as a foot note. What do lowercase letters (i.e., a) and uppercase letters (i.e., A) compare in statistical analysis? Revise to “soybased beverage (SBB)”. Be consistent with terms either “SoyaD” or “SBB”. Ok, revised. We specified the usage of different uppercase and lowercase letters in the table footnotes. (Line 403)
- Lines 337-339. Revise to “Despite the original color of A. platensis is green-blue [2], the thermal treatment leads to a degradation of pigments changing the color to a dark-green or brownish color [42]. Therefore, the impact …”. Ok, revised (Lines 415-417)
- Line 344. Revise to “as shown in the Figure 3.” Ok, revised (Line 422)
- Line 353. Are the values of the lightness on average or one representative values? It is the average difference among samples, as the increasing trend was present in the whole dataset. However, as it can be misinterpreted, we specified it in the text (please see line 431)
- Lines 361-362. Revise to “For example, Mix III in SBB showed a significant and evident decrease …”. Ok, revised (line 440)
- Line 363. Add “,” after “SSM”. Ok, revised
- Line 364. Remove “,”. Ok, revised
- Line 368-371. Revise to “Figure 3. A representative picture of skimmed milk (SSM) and soy-based beverage (SBB) inoculated with commercial mix culture I , before 368 (t0) and after 24 hours (t24) of fermentation.”. Samples were fermented with the addition of 0, 0.25, and 0.5% (v/v) lysate of Arthrospira platensis. Ok, revised (Line 450)
- Lines 376-377. Revise to “A. platensis”, “used, and on the concentration …”. Ok, revised ines 471-472
- Line 379. Revise to “… mix culture. Therefore, further …”. Sentence eliminated
The revised one slightly improved the quality of the article. However, the authors have not addressed most of my comments in the revised one (i.e., titles for Figures 2 and 3, and Tables 1, 2, and 3 are too descriptive and need to be rephrased as titles in short and statistics)
we have answered to all the comments
Reviewer 3 Report
The changes were done and it can be accepted.
Author Response
Thank for your positive opinion
Round 2
Reviewer 2 Report
The revised manuscript looks fine.
Author Response
Thank you for your positive comment